# Resistance Allele Frequency to Cry1Ab and Vip3Aa20 in *Helicoverpa zea* (Boddie) (Lepidoptera: Noctuidae) in Louisiana and Three Other Southeastern U.S. States

**DOI:** 10.3390/toxins14040270

**Published:** 2022-04-11

**Authors:** Shucong Lin, Isaac Oyediran, Ying Niu, Sebe Brown, Don Cook, Xinzhi Ni, Yan Zhang, Francis P. F. Reay-Jones, Jeng Shong Chen, Zhimou Wen, Marcelo Dimase, Fangneng Huang

**Affiliations:** 1Department of Entomology, Louisiana State University Agricultural Center, Baton Rouge, LA 70803, USA; shuconglin@tamu.edu (S.L.); yniu@agcenter.lsu.edu (Y.N.); marcelodimase@ufl.edu (M.D.); 2Syngenta Crop Protection LLC, Research Triangle Park, NC 27709, USA; isaac.oyediran@syngenta.com (I.O.); yan.zhang@syngenta.com (Y.Z.); eric.chen@syngenta.com (J.S.C.); zhimou.wen@syngenta.com (Z.W.); 3Dean Lee Research Station, Louisiana State University Agricultural Center, Alexandria, LA 71302, USA; sbrow175@utk.edu; 4Delta Research and Extension Center, Mississippi State University, Stoneville, MS 38776, USA; dcook@drec.msstate.edu; 5Crop Genetics and Breeding Research Unit, USDA-ARS, Tifton, GA 31793, USA; xinzhi.ni@usda.gov; 6Department of Plant and Environmental Sciences, Clemson University, Florence, SC 29506, USA; freayjo@clemson.edu

**Keywords:** *Helicoverpa zea*, Bt maize, F_2_ screen, resistance allele frequencies, Cry1Ab, Vip3A

## Abstract

The corn earworm/bollworm, *Helicoverpa zea* (Boddie), is a pest species that is targeted by both *Bacillus thuringiensis* (Bt) maize and cotton in the United States. Cry1Ab and Vip3Aa20 are two common Bt toxins that are expressed in transgenic maize. The objective of this study was to determine the resistance allele frequency (RAF) to Cry1Ab and Vip3Aa20 in *H. zea* populations that were collected during 2018 and 2019 from four southeastern U.S. states: Louisiana, Mississippi, Georgia, and South Carolina. By using a group-mating approach, 104 F_2_ iso-lines of *H. zea* were established from field collections with most iso-lines (85) from Louisiana. These F_2_ iso-lines were screened for resistance alleles to Cry1Ab and Vip3Aa20, respectively. There was no correlation in larval survivorship between Cry1Ab and Vip3Aa20 when the iso-lines were exposed to these two toxins. RAF to Cry1Ab maize was high (0.256) and the RAFs were similar between Louisiana and the other three states and between the two sampling years. In contrast, no functional major resistance allele (RA) that allowed resistant insects to survive on Vip3Aa20 maize was detected and the expected RAF of major RAs with 95% probability was estimated to 0 to 0.0073. However, functional minor RAs to Vip3Aa20 maize were not uncommon; the estimated RAF for minor alleles was 0.028. The results provide further evidence that field resistance to Cry1Ab maize in *H. zea* has widely occurred, while major RAs to Vip3Aa20 maize are uncommon in the southeastern U.S. region. Information that was generated from this study should be useful in resistance monitoring and refinement of resistance management strategies to preserve Vip3A susceptibility in *H. zea*.

## 1. Introduction

*Bacillus thuringiensis* (Bt) crops are genetically modified plants that express Bt toxins. Since 1996, Bt crops have been widely adopted worldwide [1,2]. Currently, more than 80% of maize, *Zea mays* L., and cotton, *Gossypium hirsutum* L., are Bt crops in the United States [3]. However, the high adoption of Bt crops has increased selection pressure and led to the evolution of resistance in pest populations. Resistance resulting in control problems in Bt maize or cotton fields has currently been observed in more than 20 cases that are associated with at least nine pest species in six countries including Argentina, Brazil, Canada, India, South Africa, and the United States [4].

The corn earworm, *Helicoverpa zea* (Boddie), is one of the most destructive agricultural pests, as well as a common species that is targeted by both Bt maize and Bt cotton in the United Sates [5,6]. Based on the mode of action, Bt toxins that are currently used in maize and cotton that have activities against *H. zea* can be categorized into three groups: Cry1A including Cry1Ab, Cry1Ac, and Cry1A.105; Cry2A including Cry2Ab and Cry2Ae; and Vip3A including Vip3a20 in maize and Vip3Aa19 in cotton. Several studies have shown that field resistance to Cry1A/Cry2A crops in *H. zea* is widely distributed in the United States [7,8,9,10,11,12,13]. The widespread Cry1A/2A resistance in the insect has become a great challenge for the sustainable use of Bt crops for pest management [14].

Cry1Ab maize was first commercialized in 1996 to control a maize borer complex in the United States and Canada [15]. Early studies reported that Cry1Ab maize could also provide some level of control for *H. zea* [16,17,18,19]. However, recent studies showed that resistance to Cry1Ab maize has occurred in many areas of the United Sates [9,11,17]. In contrast, Vip3A, an insecticidal toxin that is produced during the vegetative growth period of the *Bt* bacteria, is a relatively new insecticidal toxin that is used in Bt crops [20]. Compared to Cry toxins, Vip3A has distinct binding sites on the insect midgut brush border membrane vesicles (BBMV) [21,22]. For example, in a ligand blotting study with BBMV of *Manduca sexta*, Vip3A bound to 80-kDa and 100-kDa molecules, while Cry1Ab toxin bound to 120-kDa aminopeptidase N-like and 250-kDa cadherin-like molecules [22]. Thus, no cross-resistance between Vip3A and Cry toxins has been documented in several pest species including *H. zea* [11,12,23,24].

Transgenic cotton and maize containing Vip3A were first commercialized in the United States in 2008 and 2011, respectively, but the commercial adoption of these traits has remained limited until recently [11]. Field trials have shown that maize expressing the Vip3Aa20 toxin is highly effective against *H. zea* including the populations that are highly resistant to Cry1A/Cry2A [8]. However, due to the widespread occurrence of resistance to the Cry toxins, Vip3A becomes the only Bt toxin to provide satisfactory levels of *H. zea* control in both Bt cotton and maize. In recent years, unexpected injury (UXI) that is caused by *H. zea* in the field has become more common in Vip3A cotton and has been observed on several occasions in Vip3A maize fields as well in the southern region of the United States [14,25]. It is highly anticipated that *H. zea* will rapidly develop resistance to Vip3A if effective insecticide resistance management (IRM) plans are not implemented.

Information on Bt resistance allele frequencies (RAFs) in field populations of target pests is useful in defining susceptibility status, verifying field resistance, and mitigating resistance development [26]. During the last two decades, RAFs to Bt toxins have been investigated in many crop pest species [23,27,28,29,30,31,32,33,34,35,36,37]. One of the most common methods that is used to identify Bt resistance alleles (RAs) is the F_2_ screen [38]. Due to the difficulty in using the single-pair mating method to establish *H. zea* iso-lines, Bt RAFs in *H. zea* have been investigated in only three studies so far. Burd et al. [39] used an F_1_ screen and estimated the non-recessive Cry1Ac and Cry2Ab RAF in *H. zea* in North Carolina. Recently, Yang et al. [37] and González et al. [10] utilized an F_2_ screen and examined RAFs to Cry1Ac, Cry2Ab, and a Vip3A variant Vip3Aa39 in Texas populations. RAFs to all other Bt toxins in *H. zea* remain unknown. In this study, we used a group-mating approach and successfully established 104 F_2_ iso-lines of *H. zea* from field-collected larvae during 2018 and 2019 from four southeastern states of the United States. The majority of the iso-lines were derived from field collections in Louisiana (LA), followed by Mississippi (MS), Georgia (GA), and South Carolina (SC). The F_2_ screens showed that RAs in the *H. zea* populations to Cry1Ab maize were abundant, while functional major RAs to Vip3Aa20 maize were not detected. 

## 2. Results

### 2.1. Establishment of 104 H. zea F_2_ Iso-Lines for Screening Cry1Ab and Vip3Aa20 Resistance Alleles

A total of 104 F_2_ iso-lines were successfully established for *H. zea* that were collected from eight locations in LA, MS, GA, and SC during 2018 and 2019 by using a group-mating method. Among these, 52 iso-lines were established in each year. Among the 52 lines that were established in 2018, 33 lines were from LA (Winnsboro—27 lines, Baton Rouge—5, St. Joseph—1); 11 lines were from MS (Stoneville—7, Leland—4); 6 lines were from Tifton, GA; and 2 lines were from Florence, SC. All 52 lines that were established in 2019 were collected from Alexandria, LA. Of the 11 lines from MS, three were generated from larvae that were collected from VT Double PRO maize plants expressing Cry1A.105 and Cry2Ab2, while all other 101 lines were established from larvae that were collected from non-Bt plants. F_2_ neonates of 94 of the 104 lines were screened against both Cry1Ab at 10.0 µg/cm^2^ and Vip3Aa20 at 5 µg/cm^2^. For the other 10 iso-lines, three and seven lines were tested only against Cry1Ab and Vip3Aa20, respectively. To facilitate data analysis, the 104 iso-lines were organized into three groups: one from LA in 2018 (33 lines), one from LA in 2019 (52 lines), and one from the other three states in 2018 (19 lines) (Table 1).

### 2.2. A Total of 87 of the 97 H. zea Iso-Lines Screened against Cry1Ab Were Identified as Potential Positive Lines Possessing Resistance Alleles to the Toxin

Overall, the *H. zea* iso-lines survived well in F_2_ screen against Cry1Ab, and the larval survival rates were similar among the three insect groups. Among the 11,815 F_2_ neonates of the 97 iso-female lines (in most cases, 128 neonates/line) that were screened against Cry1Ab, 3816 individuals (or a survivorship rate of 32.3%) in 95 lines survived the seven-day F_2_ screen, which included 2204 late-stage larvae (≥3rd instars) (18.7%) in 87 lines (Table 1). Based on our previous studies [8,11], *H. zea* neonates that survive and develop to the late-stage larvae when exposed to 10 µg/cm^2^ of Cry1Ab may carry RAs to the Bt toxin. Thus, the 87 iso-lines that survived and had larvae that developed to late-stage larvae in F_2_ screen were considered as potential positive lines (PPLs) carrying RAs to Cry1Ab. In detail, for the 32 iso-lines that were established from the 2018 LA collections, 31 lines had surviving larvae after seven days in the F2 screen with a total of 1448 survivors (or a survivorship rate of 35.4%), which included 571 late-stage larvae (13.9%) in 26 lines (or 26 PPLs) (Table 1). All 18 lines that were collected during 2018 from the other three states survived the seven-day screen with a total of 691 live larvae (30.0%) including 334 late-stage larvae (14.5%) in 16 PPLs. For the 47 lines that were collected from LA during 2019, 46 lines survived the seven-day screen with a total of 1677 survivors (31.0%) including 1299 late-stage larvae (24.0%) in 45 PPLs (Table 1).

### 2.3. A Total of 10 of the 101 H. zea Iso-Lines Screened against Vip3Aa20 Were Identified as Potential Positive Lines Possessing Resistance Alleles to the Toxin

The larval survival rates of the *H. zea* iso-lines in the Vip3Aa20 F_2_ screen were considerably lower than those that were observed in the screen against Cry1Ab. Among the 12,148 larvae of the 101 iso-lines (in most cases 128 neonates/line) that were screened against Vip3Aa20 toxin, 256 individuals (or a survivorship rate of 2.1%) in 39 lines survived, which included 52 late-stage larvae (0.43%) from 10 lines (Table 1). According to our previous study in the reference [11], *H. zea* neonates that survive and grow to late-stage larvae (≥3rd instars) at 5 µg/cm^2^ of Vip3Aa20 may carry RAs to the toxin. Thus, the 10 iso-female lines that survived and had larvae that reached the late-stage larvae in the F_2_ screen were considered as PPLs for Vip3Aa20. More specifically, in the 33 iso-lines that were collected in 2018 from LA, 24 lines survived after seven days in the F_2_ screen with 182 survivors (4.31%) including 34 late-stage larvae (0.80%) in 6 PPLs (Table 1). Among the 19 lines that were collected in 2018 from the other three states, 9 lines survived the seven-day screen with 41 live larvae (1.7%) including six late-stage larvae (0.025%) in 1 PPL. For the 49 lines that were collected in 2019 from LA, six lines survived the seven-day screen with 33 survivors (0.60%) including 12 late-stage larvae (0.22%) from 3 PPLs (Table 1).

### 2.4. No Positive Correlations in Larval Survivorship for the Iso-Lines Screened against Cry1Ab and Vip3Aa20

Linear regression analysis indicated that for the 94 iso-lines that were tested against both toxins, there was no significant relationship between larval survivorship on Cry1Ab and that on Vip3Aa20. The correlation coefficient (*R*) for larval survivorship on Cry1Ab (independent variable X) and Vip3Aa20 (dependent variable Y) diet was 0.098 (Y = 0.0069 − 0.010 X; *R*^2^ = 0.0096, *t* = −0.94, df = 1, *p* = 0.3475).

### 2.5. Offspring of the Potential Positive Lines against Cry1Ab Exhibited Significant Resistance Levels to the Toxin

Three PPL strains (PPL-Cry1Ab-2018-I, PPL-Cry1Ab-2018-II, and PPL-Cry1Ab-2018-III) of *H. zea* were established from the survivors of the 42 PPLs that were identified in the F_2_ screen against Cry1Ab during 2018. One strain (PPL-Cry1Ab-2019) was developed from the 46 PPLs that were identified during 2019. Results from the diet over-lay bioassays demonstrated that all four PPL strains were significantly resistant to Cry1Ab. The bioassay that was conducted in 2018 showed that the known susceptible strain (SS-BZ) had an LC_50_ of 0.29 µg/cm^2^ (Table 2). PPL-Cry1Ab-2018-I, which was selected with 10 µg/cm^2^ of Cry1Ab before the current bioassay, had the highest resistance ratio (>109-fold relative to SS-BZ) to Cry1Ab compared to SS-BZ (Table 2). The LC_50_ of PPL-Cry1Ab-2018-II (3.31 µg/cm^2^) was significantly greater than that of SS-BZ based on their non-overlapped 95% CIs (11-fold difference). Similarly, the LC_50_ (8.61 µg/cm^2^) for PPL-Cry1Ab-2018-III was also significantly higher than that of SS-BZ (30-fold difference). In the bioassays that were conducted during 2019, SS-BZ had an LC_50_ of 0.14 µg/cm^2^, while PPL-Cry1Ab-2019 had an LC_50_ of 13.76 µg/cm^2^ and the 98-fold difference was significant (Table 2).

### 2.6. Offspring of the Potential Positive Lines against Vip3Aa20 Exhibited Low Levels of Resistance to the Toxin

Two insect strains (PPL-Vip3Aa20-2018-I and PPL-Vip3Aa20-2018-II) were established from the survivors of the seven PPLs that were identified in an F_2_ screen during 2018, and one strain (PPL-Vip3Aa20-2019) from the three PPLs during 2019. Diet overlay bioassays with Vip3Aa20 showed that all three PPL strains had significantly greater LC_50_s compared to SS-BZ, but the resistance ratios were much lower than those that were observed for the Cry1Ab PPL strains. SS-BZ, that was assayed in 2018, showed an LC_50_ of 0.36 µg/cm^2^. In contrast, PPL-Vip3Aa20-2018-I, which had an additional selection with 3.16 µg/cm^2^ Vip3Aa20 before being tested in the current bioassay, had an LC_50_ of 3.96 µg/cm^2^ which was significantly greater than that for SS-BZ based on the non-overlapped of the 95% CIs (11-fold difference). The LC_50_ value for PPL-Vip3Aa20-2018-II was 2.51 µg/cm^2^, which was also significantly greater than that for SS-BZ (seven-fold difference). The LC_50_ for SS-BZ during 2019 was 0.47 µg/cm^2^, while the LC_50_ for PPL-Vip3Aa20-2019 was 1.89 µg/cm^2^, and the four-fold difference was also significant.

### 2.7. Cry1Ab PPL Strains Completed Larval Development on Cry1Ab Maize Ears, While the Vip3Aa20 PPLs Could Not Survive on Vip3Aa20 Maize Ears

Detached maize ear assays showed that *H. zea* neonates survived well on non-Bt maize ears. With an infestation rate at 10 SS-BZ neonates/ear for a 22-day infestation period, two live larvae (one fifth and one sixth instars) and one pupa were recovered from five non-Bt ears. From 10 Cry1Ab maize ears that were infested with 100 neonates of PPL-Cry1Ab-2018-I (10 neonates/ear) for 22 days, one fifth instar, one sixth instar, and one pupa were recovered after 22 days. Similarly, from five Cry1Ab ears that were infested with 100 neonates (20 neonates/ear) of PPL-Cry1Ab-2019 for 22 days, four live larvae (fourth to sixth instars) and two pupae were recovered. These assays on Cry1Ab maize ears showed that the PPLs possessed functional major RAs that allowed the resistant insects to survive and complete larval development on the Cry1Ab maize ears [40,41,42]. In contrast, no survivors were recovered from the 22 Vip3Aa20 maize ears that were infested with 330 neonates (15 neonates/ear) of PPL-Vip3Aa20-2018-I after 14 days. Similarly, no survivors were observed from 10 Vip3Aa20 ears that were infested with 100 neonates (10/ear) of PPL-Vip3Aa20-2019 after 10 days. The maize ear assays showed that the PPLs that were identified in the F_2_ screen with Vip3Aa20 did not carry functional major RAs that could allow the resistant individuals to survive and complete larval development on Vip3Aa20 maize ears.

### 2.8. Resistance Allele Frequencies to Cry1Ab Were High in the H. zea Populations

Baseline assays showed that a mixed strain that was derived from the F_1_ progeny from multiple *H. zea* iso-lines survived well on non-Bt diet with an average seven-day survivorship of 98.4 ± 1.2% (mean ± sem). On the diet that was treated with 10 µg/cm^2^ of Cry1Ab, 100% mortality of larvae from a known susceptible strain (SS-BZ) was observed after seven days, while a known resistant strain (Cry1Ab-RR) had a seven-day survivorship of 93.8 ± 3.4%, and the survivorship of F_1_ hybrid (Cry1Ab-RS) of SS-BZ and Cry1Ab-RR was 57.0 ± 2.3%. Based on the Mendelian inheritance models as described in the reference [33], the expected number of survivors of an iso-line in the F_2_ screen was estimated by using the equation: expected number of survivors = number of F_2_ neonates of an iso-line screened × (0 × f_SS_ + 0.570 × f_RS_ + 0.938 × f_RR_) (Table 3). Here, f_SS_, f_RS_, and f_RR_ represent the expected SS, RS, and RR genotype frequencies, respectively, in the F_2_ progeny of an iso-line.

RAF to Cry1Ab were high in the populations of *H. zea* that were collected from the four states. Chi-square tests showed that among the 32 LA iso-lines that were screened against Cry1Ab in 2018, only 6 lines did not carry RAs to Cry1Ab, 24 lines carried 1 RA, and 2 lines possessed 2 RAs. Thus, a total of 28 RAs were identified in the 32 iso-lines, and the expected RAF to Cry1Ab in the population based on a Bayesian analysis [33] was estimated to be 0.223 (Table 4). In the 18 iso-lines that were established in 2018 from the other three states, a total of 16 RAs were detected, in which 2 iso-lines had zero RA and 16 possessed 1 RA. The Cry1Ab RAF of the population from the three states was estimated to be 0.230, which was not significantly different from the RFA of the LA population based on the overlap of 95% CIs. Thus, the RAF for the combined 50 iso-lines that were screened against Cry1Ab during 2018 from the four states was 0.223 with a 95% CI of 0.156 to 0.298.

Among the 47 iso-lines that were screened from LA in 2019, a total of 55 RAs were found, in which two iso-lines had zero RA to Cry1Ab, 37 iso-lines carried 1 RA, 6 iso-lines possessed 2 RAs, and 2 iso-lines had 3 RAs. The expected Cry1Ab RAF in the population was estimated as 0.295, which was also not significantly different compared to the RAFs that were observed during 2018 based on overlapped of 95% CIs (Table 4). Altogether, 99 RAs were identified in the 97 iso-lines from the four states during the two years, and the overall RAF to Cry1Ab was estimated to be 0.256 with a 95% CI of 0.214 to 0.301 (Table 4).

### 2.9. Resistance Allele Frequencies to Vip3Aa20 in the H. zea Populations Were Still Low for Functional Major Alleles, but Were Not Uncommon for Functional Minor Alleles

The functional major RAs to Vip3Aa20 were uncommon in the *H. zea* populations that were collected from the four states. Based on the performance of the progeny of the F_2_ survivors on Vip3Aa20 maize ears, none of the 101 iso-lines that were screened from the four states during 2018 and 2019 possessed functional major RAs that allowed the resistant larvae to survive on Vip3Aa20 maize ears. By using Bayesian analysis [43,44], the estimated RAF with 95% probability for the overall *H. zea* populations was 0 to 0.0073 (Table 4).

The results of the diet over-lay bioassays demonstrated that all three strains that were derived from the F_2_ survivors had low, but significant levels of resistance to the Vip3Aa20 (Table 2). Thus, the 10 PPLs that survived the seven-day F_2_ screen with late-stage larvae, but could not survive on Vip3Aa20 maize ears, were considered to possess functional minor RAs to Vip3Aa20 maize. More specifically, six of the 33 iso-line from LA during 2018 were identified to possess functional minor RAs to Vip3Aa20 based on the seven-day survival in F_2_ screen, and thus the RAF of minor alleles to Vip3Aa20 maize was estimated to be 0.054. For the 19 iso-lines from the three other states, one line was considered to carry a minor RA with an estimated RAF of 0.025 (Table 4). For the 49 iso-lines from LA during 2019, three iso-lines possessed minor RAs to Vip3Aa20 with a combined RAF of 0.020 (Table 4). The estimated RAFs for the three populations were not significantly different based on their overlapped 95% CIs. Across all the populations and states, the RAF of functional minor alleles to Vip3Aa20 maize was estimated to be 0.028 with a 95% CI of 0.014 to 0.047 (Table 4). The results showed that functional minor Vip3Aa20 RAs in the *H. zea* populations were not uncommon.

## 3. Discussion

As a preparation of the current F_2_ screen, we attempted to establish two-parent families of *H. zea* from field collections by using a single pair-mating method that has been commonly used with other moth species [30,33]. These efforts were unsuccessful due to the mating difficulty of the single pairing. Therefore, we used the group-mating approach to establish the necessary *H. zea* iso-lines for the F_2_ screen in this study. A main potential limitation of the use of group-mating is that it may result in a female mating with more than one male, and vice versa, which could cause errors in the RAF estimations. The F_2_ iso-lines that were established using group-mating of females that were derived from field-collected larvae with known laboratory susceptible males [10] could preclude this limitation. However, the related labor costs would increase drastically because only a single feral parent in each F_2_ iso-line can be evaluated. In addition, feral males are not assessed with the group-crossing between feral females and known susceptible males. Previous studies have shown that most *H. zea* females mate only once in natural environments [45]. In another related study, we dissected the laboratory-mated *H. zea* females after oviposition to record the number of spermatophores in the bursa copulatrix. Among 110 females that were dissected, 92.7% females contained a single spermatophore (W. Yu, F. Huang unpublished data). The 110 females that were dissected included most females that successfully produced viable offspring for the current F_2_ screen. The dissections indicated that *H. zea* females typically also mated only once in the laboratory. Additionally, for multiple-mated female *Heliothis virescens* (F.), a species that was closely related to *H. zea*, offspring were typically derived from a single male [46]. Thus, the chance of a mated female parent of a *H. zea* iso-line passing genetic materials from more than one male to the offspring should be rare. Furthermore, our observations also showed that even with the group-mating approach, the success rate of F_2_ iso-line establishment from the individual females was low (only 5–10%; data not shown), suggesting that the probability of a male successfully mating and transferring its genetic materials to more than one female was also low. Therefore, the likelihood of more than four alleles at a locus for each female parent in our F_2_ screen should be low and the effect, if any, should not cause a significant error in the RAF estimations.

Two decades ago, *H. zea* in the U.S. was susceptible to Cry1Ab [19]. However, Dively et al. [7] reported that Cry1Ab maize ears that were damaged by *H. zea* in Maryland increased from 6.3% in 1996 to 85.1% in 2016. Similarly, field trials in North and South Carolina during 2012–2013 also showed that Cry1Ab maize no-longer inhibited the development of *H. zea* [47]. The results of these studies suggest that *H. zea* has already developed resistance to Cry1Ab maize in the U. S. Recently, Niu et al. [11] reported that Cry1Ab resistance in *H. zea* is widely distributed in the southeastern U.S. region; this finding was further validated by the high RAF (0.254) to Cry1Ab maize that was detected in the current study. In addition to the long-term and widespread use of the single-gene Cry1Ab maize, the extensive planting of Cry1Ac cotton is likely also a main factor for the observed Cry1Ab resistance in *H. zea* because the two toxins share the same mode of action [48,49]. The prior selection from the widespread planting of Cry1A maize and cotton is also believed to be an important factor contributing to the rapid evolution of resistance to Cry1A.105 maize in the U.S. [13].

Compared to the high Cry1Ab RAF, functional major RAs that allowed resistant individuals to survive on Vip3Aa20 maize was not detected in the 101 iso-lines that were screened in the current study, suggesting that RAF to Vip3Aa20 is still low in the *H. zea* populations from the four states. A previous F_2_ screen detected two out of 114 F_2_ iso-lines of a Texas *H. zea* population that were collected in 2019 with RAs to Vip3Aa39 [37]. The Vip3Aa39 RAF in the Texas population was estimated to be 0.0065 [37]. Vip3Aa39 is a similar toxin to the Vip3Aa20 that is expressed in maize, but the survival of the Vip3Aa39-resistant *H. zea* on Vip3A maize/cotton remains unknown. To date, resistance leading to field control problems of Vip3A maize and cotton has not yet been documented in *H. zea*. Multiple field trials in Maryland, Minnesota, Texas, Louisiana, and other states all reported that maize traits containing Vip3Aa20 were still effective for *H. zea* control [7,8,25,50,51]. A recent study in the reference [11] showed that 29 *H. zea* populations that were collected from the southeastern U.S. region in 2018 and 2019 were susceptible to Vip3Aa20. Besides the relative short commercial use of Vip3A crops, another major factor that has preserved the Vip3A susceptibility is the lack of cross-resistance between Vip3A and Cry toxins [52,53,54]. The lack of correlation between larval survivorship of the F_2_ lines on Cry1Ab and Vip3Aa20 toxins that was observed in the current study provides further evidence that there is no cross-resistance between Cry1Ab and Vip3A toxins in *H. zea*.

While more studies are warranted to validate if the RAF in *H. zea* to Vip3A meets the rare RA requirement as defined in the ‘high/dose-refuge’ IRM strategy [15,41], the results of the current study showed that functional minor RAs to Vip3Aa20 maize are not uncommon (RAF = 0.028). Although insects carrying such functional minor RAs could not complete their larval development on Bt plants expressing Vip3Aa20 in the no-choice laboratory bioassay, the constant presence of such individuals in the landscape may still have important implications for resistance evolution. Factors such as larval movement [55,56,57,58], seed blends [59], polygenetic inheritance, varied susceptibility among larval stages [60,61], different expression levels of insecticidal toxins in different plant tissues and across plant growth stages [62,63,64], potential antagonism among Bt toxins [65], as well as, the effect of abiotic environmental factors on Bt toxin expression [62] can create favorable conditions for the selection of functional minor RAs and thus, accelerate the development of resistance. As mentioned above, UXIs of *H. zea* on both Vip3A cotton and maize have been observed in several incidents in the U.S. [14,25]. Field populations of *H. zea* possessing minor RAs could also be a factor for the observed UXIs. For example, during the 2021 crop season, >30 maize ears expressing Vip3Aa20 toxin with the second and third instar *H. zea* larvae were collected from two UXI field plots in Louisiana. The larvae were placed in the laboratory rearing facility, but after three weeks none of the larvae developed to ≥fifth instars (F. Huang, unpublished). The results indicated that functional minor RAs might be associated with the observed UXIs under some specific circumstances. Nevertheless, the roles of such functional minor RAs in evolution of Bt resistance are not fully understood yet and more detailed studies are warranted.

## 4. Conclusions

RAs to Cry1Ab maize in *H. zea* are common in the southeastern region of the United States, further validating that field resistance to Cry1Ab toxin in field populations of the insect has widely occurred in the regions. In contrast, RAF of functional major RAs to Vip3Aa20 maize in *H. zea* is low, but additional studies are needed to validate if the RAF meets the rare RA requirement as defined in the ‘high/dose-refuge’ strategy. In addition, the F_2_ screen also showed that functional minor Vip3Aa20 RAs in *H. zea* are not uncommon. The detection of minor RAs may have important implications in the evolution of resistance. The F_2_ screen showed no positive relationship between larval survivorship of the *H. zea* iso-lines that were exposed to Cry1Ab and Vip3Aa20. The results present further evidence that there is no cross-resistance between the two Bt toxins. Information that was generated from this study should be useful in resistance monitoring and the development of IRM plans to preserve Vip3A susceptibility in field populations of *H. zea*.

## 5. Materials and Methods

### 5.1. Insect Collections and Rearing

During the 2018–2019 cropping seasons, 2nd to 6th instars of *H. zea* were collected from maize fields at eight sampling sites in four southeastern U.S. states, including four sites in LA, two sites in MS, one site each in GA and SC. The four LA sampling sites were located near Baton Rouge, Alexandria, Winnsboro, and St. Joseph; the two MS sites were near Leland and Stoneville; the GA site was near Tifton, and the SC site was near Florence. The field-collected larvae were individually reared in 30 mL plastic cups (SOLO, Chicago, IL, USA), each containing ~8 g of a meridic diet (Ward’s *Heliothis* diet, Rochester, NY, USA). The insect rearing cups were held in 30-well trays (Bio-Serv, Frenchtown, NJ, USA) and the trays were placed at 20–26 °C until pupation. Varied temperatures were used to synchronize the development of the field-collected insects.

### 5.2. Establishment of F_1_ and F_2_ Iso-Lines

A method of single-pair mating has been widely used to establish two-parent iso-lines of Lepidopteran species for F_2_ screens to detect Bt RAs [23,30,33,36,66]. Prior to this study, several attempts were made to use the similar single-pair mating method to establish the necessary two-parent families of *H. zea* from field collections, but all failed. Thus, in this study, a group-mating approach was utilized to establish iso-lines. For group-mating, mature pupae that were close to emerging were collected from the above-mentioned insect rearing cups and placed into plastic containers (10-cm diameter and 2-cm height). The plastic containers each holding ≤ 80 pupae of a population (depending on the number of pupae that were available) were placed in Seville Classic 20-L cages (Torrance, CA, USA) each containing approximately 200 g of vermiculate (Sun Gro, Pine Bluff, AR, USA). A cup containing a paper towel that was saturated with 10% honey-water solution was placed in the center base of cage to supply water and nutrition for the adults [67]. The open top of the cage was then covered with gauze. Each cage had one to three plastic containers containing pupae, depending on the total number of pupae that were available in a population. The cages with pupae were arranged in an insect rearing room at 26 °C, >70% humidity, and a 14:10 h (L:D) photoperiod for adult emergence [11].

When a certain number (in most cases 20–60 adults/cage) of adults had emerged, the plastic containers containing the pupae inside the cages that had not emerged were removed from the cages and transferred into new identical 20-L cages for continued adult emergence. The adult cages containing the newly emerged adults were maintained in the same insect rearing room and were allowed to group-mate for 2 to 3 days after emergence. After group-mating, the females were individually transferred into 3.8-L paper containers (1 female/container) (Huhtamaki Foodservice, DeSoto, KS, USA). Each paper container contained ~50 g vermiculate and a cup containing a paper towel that was saturated with 10% honey-water solution, and the tops of the paper containers were covered using nonsterile gauze nets (Fisher Scientific, Pittsburgh, PA, USA). The paper containers each containing a single female were placed in the above-mentioned insect rearing room. The offspring that were produced from each female were considered as an iso-line.

F_1_ eggs that were produced from each iso-line were collected daily. A total of 60–120 F_1_ neonates of each iso- line were reared on the same meridic diet that is mentioned above in 128-cell trays (1 larva/cell) (Bio-Ba-128, C-D International Inc., Pitman, NJ, USA). After 5–7 days, the larvae in the cells were moved to 30-mL cups and continued to be reared under the same conditions that are described above until the pupal stage. The F_1_ pupae from each iso-line were placed in 20-L mesh cages (one cage per iso-line) for sib-mating within each iso-line. For each iso-line, the F_2_ eggs were collected daily and placed in chambers for egg hatching.

### 5.3. F_2_ Screening Procedures

Based on our previous study [11], the diet over-laid with Cry1Ab toxin at 10 μg/cm^2^ and Vip3Aa20 at 5 μg/cm^2^ were used as the diagnostic doses in the F_2_ screen to detect the corresponding RAs. Lyophilized Cry1Ab (84.3% purity) and Vip3Aa20 (86.5% purity) toxins used in F_2_ screen were provided by Syngenta Biotechnology (TRC, Research Triangle NC, USA) [11,24]. The F_2_ screen was conducted in the 128-cell C-D International trays that were mentioned previously. To prepare the diet, ~0.8 mL of the Southland meridic diet (Lake Village, AR, USA) was placed in each cell with BD 20- or 30-mL Luer-Lok^TM^ syringes (BD, Franklin Lakes, NJ, USA). Appropriate amounts of lyophilized Cry1Ab and Vip3Aa20 toxins were diluted in a 0.1% Triton-X100 solution. For each Bt toxin solution, 50 µL of the diluted solution was added to the surface of the diet in each cell using Eppendorf Repeater^®^ M4 pipettes (Pipett.com, San Diego, CA, USA). After the Bt solution dried, one newly hatched larva of an iso-line was placed on the surface of the Bt-treated diet. In most cases (176 of 198 screens), for each iso-line and Bt toxin, 128 F_2_ neonates (one tray) were screened. For the other lines (22 screens), the number of F_2_ neonates that were screened per iso-line ranged from 60 to 96 based on the availability of F_2_ neonates. The F_2_ screen trays were arranged in environmental chambers at 26 °C, ~50% r.h., and a 16:8 h (L:D) photoperiod. The number of stunted larvae (1st and 2nd instars), and normally developed larvae (i.e., late-stage larvae or ≥3rd instars) for each iso-line were recorded after 7 days in the F_2_ screen. The iso-lines with surviving late-stage larvae after 7 days in the F_2_ screen were considered as PPLs that possessed RAs to Cry1Ab or Vip3Aa20. The expected survivorship of the F_2_ progeny for each iso-line in the F_2_ screen was estimated by the method that was described in the reference [33]. Mathematically, 6.25% of the F_2_ larvae would survive in the F_2_ screen if a recessive RA to the Bt toxin existed in one of the two parents of the iso-line [38].

### 5.4. Susceptibility of the Offspring of PPLs to Cry1Ab and Vip3Aa20 Toxins

To establish insect strains for resistance confirmation, the survivors from multiple PPLs that were screened at a similar time were mixed. The PPL strains were then used in dose-response bioassays to confirm if the PPLs were resistant to Cry1Ab or Vip3Aa20. With the iso-lines established from insects that were collected in 2018, three PPL strains were established from the F_2_ survivors against Cry1Ab: PPL-Cry1Ab-2018-I, PPL-Cry1Ab-2018-II, and PPL-Cry1Ab-2018-III, and two strains were established from the survivors against Vip3Aa20: PPL-Vip3Aa20-2018-I and PPL-Vip3Aa20-2018-II. For the insects that were collected in 2019, one PPL strain (PPL-Cry1Ab-2019) was established from the F_2_ survivors against Cry1Ab and one (PPL-Vip3Aa20-2019) was established from the survivors against Vip3Aa20. As many larvae survived in the F_2_ screen against Cry1Ab, only late-stage survivors were used to generate the four Cry1Ab-related strains. As the number of survivors in the F2 screen against Vip3A20 was limited, all the live larvae (small and late-stage larvae) in the 10 PPLs were used to generate the three Vip3Aa20-related strains to ensure colony establishment. One additional selection with 10 µg/cm^2^ of Cry1Ab for PPL-Cry1Ab-2008-I and 5 µg/cm^2^ of Vip3Aa20 for PPL-Vip3Aa20-2008-I was performed before the offspring of the two reselected strains were used in resistance confirmation. The progenies of all other strains were directly used in diet over-lay bioassays without additional selection as described below.

Susceptibilities of the above-mentioned seven PPL strains that were generated from the survivors of the F_2_ screen, along with a known Bt-susceptible strain (SS-BZ), were assayed using a diet over-lay method as described in the reference [11]. SS-BZ, provided by Benzon Research Inc. (Carlisle, PA, USA), has previously been shown to be susceptible to Cry1Ab and Vip3A toxins [8,11,24]. The same Cry1Ab and Vip3Aa20 toxins that were used in the F_2_ screen were utilized in the diet over-lay bioassays, which were conducted in 128-cell C-D International trays. Each bioassay consisted of a series of Bt concentrations (e.g., 0.01, 0.0316, 0.10, 0.316, 1.0, 3.16, 10.0, and up to 31.6 μg/cm^2^) plus a negative control that contained 0.1% Triton-X100 only. The procedures of the Bt diet for the bioassays were the same as described in the preparation of the diagnostic doses for the F_2_ screen. One neonate (<24 h old) of an insect strain was placed on the diet surface in each cell of the 128-cell trays. Each concentration or control in a bioassay consisted of four replications and each replication had 16–32 neonates per concentration. The bioassay trays with neonates were kept in environmental chambers at 26 °C, ~50% r.h. and 16:8 h (L:D). The number of dead larvae and the number of live larvae that had not developed past the 2nd instar were recorded after 7 days.

To determine Bt susceptibility, the larval mortality for each treatment replication was measured as practical mortality, which was computed based on the sum of the number of dead and live larvae (<3rd instar) after 7 days in the bioassays divided by the total number of individuals that were assayed in the treatment replication [68]. The practical mortality for each treatment replication was then corrected based on the control using the method of the reference [69]. The corrected larval mortality was analyzed using probit models [70] with SAS PROC PROBIT [71] to calculate the LC_50_ and associated 95% CI for SS-BZ and the insect strains that were derived from survivors in the F_2_ screen. The resistance ratio of a PPL strain was calculated by dividing LC_50_ value of the strain by the LC_50_ of BZ-SS. The resistance ratio was used to determine if a PPL strain, derived from F_2_ survivors, possessed Cry1Ab or Vip3Aa20 RAs.

### 5.5. Resistance Reconfirmation on Bt Maize Ears

The survival of the PPL-Cry1Ab-2018-I, PPL-Vip3Aa20-2018-I, PPL-Cry1Ab-2019, and PPL-Vip3Aa20-2019 along with SS-BZ were also evaluated on non-Bt and Bt maize ears in the laboratory to reconfirm if the resistance of the PPLs was functional. The seeds of maize iso-lines expressing Cry1Ab (Bt11 event) or Vip3Aa20 (MIR162 event) and a non-Bt maize line were provided by Syngenta Crop Protection and planted in the greenhouse in Baton Rouge, LA, USA. At the R2 plant stage, the ears along with silks, husks, and shanks were removed from plants as described in the reference [24]. An ELISA technique (Quantiplate™ kits; EnviroLogix, Portland, ME, USA) was used to confirm the expression/non-expression of the Bt toxins in the leaf tissue of the greenhouse-grown plants.

The removed ears from the greenhouse-grown plants were manually infested with 10, 15, or 20 neonates (<24 h old) of a *H. zea* strain on the silks of each ear, depending on the number of neonates that were available and number of ears that were assayed. The maize ears containing neonates were placed in 5.7-L plastic boxes with 2–3 pieces of paper towel on the bottom of the box for absorbing extra moisture. Each box contained 1–2 maize ears that were also separated with paper towel. There were 5 to 22 maize ears for each test. The boxes containing maize were placed in an insect rearing room at 24–26 °C, ~50% relative humidity, and 16:8 h (L:D). The ears were replaced with fresh ears when necessary. The larval survival and pupation on the ears were monitored for 10 or 22 days, depending on the survival status. As described in the references [30,31], if the larvae of a PPL survived to the pupal stage on Bt maize ears while those from the susceptible strain did not, the PPL strain was considered to carry functional major RAs to the Bt toxin in maize ears. Otherwise, if a PPL strain showed some levels of resistance in the dose-response bioassay, but none of the individuals survived to the pupal stage (or mature larvae) on the Bt maize ears, the PPL strain was considered to carry functional minor RAs to the Bt toxin.

### 5.6. Estimation of Resistance Allele Frequencies to Cry1Ab and Vip3Aa20

All four PPLs showed significant levels of resistance to Cry1Ab toxin in diet over-lay bioassays and both PPLs that were tested on Cry1Ab maize ears had larvae that survived and developed to the pupal stage (see Section 2). Thus, all PPLs that were identified in the Cry1Ab F_2_ screen were considered to carry functional major RAs to Cry1Ab maize. In addition, the survival rate that was observed in the Cry1Ab F_2_ screen was high (e.g., much greater than 6.15%) for many iso-lines (see Table 2), suggesting that the parents of some iso-lines possessed more than one Cry1Ab RA. To determine the number of Cry1Ab RAs in the parents of a *H. zea* iso-line, Chi-square tests were conducted to compare the observed number of survivors in a line in the F_2_ screen and the expected number of survivors based on 0-, 1-, 2-, 3-, and 4-allele Mendelian inheritance models as described in the reference [33].

The expected number of survivors for an iso-line in the F_2_ screen was estimated based on the baseline survivorship of the above-mentioned susceptible strain (SS-BZ), a known homozygous Cry1Ab-resistant strain (Cry1Ab-RR) and an F_1_ heterozygous Cry1Ab resistant genotype (Cry1Ab-RS). The Cry1Ab-RR strain was developed from iso-lines of field populations that were collected from non-Bt maize fields during 2019 in LA that showed resistance to Cry1Ab, but susceptible to Cry2Ab2 in an F_2_ screen. The Cry1Ab-RR has been shown to be highly resistant to purified Cry1Ab toxin and maize plants expressing a single Cry1Ab toxin. The Cry1Ab-RS was the F_1_ hybrid of reciprocal crosses between Cry1Ab-RR and BZ-SS. The baseline survivorships of BZ-SS, Cry1Ab-RR, and Cry1Ab-RS were obtained by using the diet over-lay assay at 10 μg/cm^2^ Cry1Ab toxin. At the same time, baseline survival of a mixed F_1_ population from multiple iso-lines that were derived from the field collections in 2019 was also determined on non-Bt diet. In the assay at 10 μg/cm^2^ Cry1Ab toxin, there were four replications for each insect strain and 32 larvae in each replication (*n* = 128). In the assay with non-Bt diet, there were three replications with 128 larvae per replication (*n* = 384). The larval survival in the baseline assays was recorded after 7 days as described previously in the F_2_ screen. The number of RAs in the mated female parent of each iso-lines was determined based on the baseline survival data and Chi-square tests at α = 0.05.

As survivors were rare (see Section 2) in the Vip3Aa20 F_2_ screen, multiple Vip3Aa20 RAs in the mated female parent of an iso-line were unlikely. Thus, if an iso-line had survival in the Vip3Aa20 F_2_ screen, the mated female was considered to possess only a single RA. The expected RAF and the corresponding 95% CIs for the *H. zea* populations were calculated using Bayesian analysis for Vip3Aa20 [43,44], while the RAF values and 95% CIs for Cry1Ab were estimated with the method described in the reference [33].

## Figures and Tables

**Table 1 toxins-14-00270-t001:** Larval survival of *H. zea* iso-lines that were collected in Louisiana and other states in F_2_ screen for detecting Cry1Ab and Vip3Aa20 resistance alleles ^a^.

Year	State	F_2_ Screen Against Cry1Ab						F_2_ Screen Against Vip3Aa20					
		Total No. Lines Screened	Total Neonates Screened	Total No. Lines Survived	Total No. Larvae Survived	No. Lines Having ≥3rd Instars	No. ≥3rd Instars	Total No. Lines Screened	Total Neonates Screened	Total No. Lines Survived	Total No. Larvae Survived	No. Lines Having ≥3rd Instars	No. ≥3rd Instars
2018	Louisiana	32	4096	31	1448	26	571	33	4224	24	182	6	34
	Other states	18	2304	18	691	16	334	19	2432	9	41	1	6
Sub-total (2018)		50	6400	49	2139	42	905	52	6656	33	223	7	40
2019	Louisiana	47	5415	46	1677	45	1299	49	5492	6	33	3	12
Total		97	11,815	95	3816	87	2204	101	12,148	39	256	10	52

^a^ Number of F_2_ neonates per iso-line that were screened against Cry1Ab was 60–96 for each of 13 iso-lines that were collected in 2019 and against Vip3Aa20 was 64–84 for each of 9 iso-lines that were collected during 2019. For all other iso-lines, the number of F_2_ neonates that were screened per iso-line was 128 for each Bt toxin.

**Table 2 toxins-14-00270-t002:** Susceptibility of the offspring of *H. zea* survivors in the F_2_ screen against Cry1Ab and Vip3Aa20 toxins.

Strains	No. Neonates Assayed	Slope ± SE	LC_50_ (95%CI, or Larval Mortality at the Highest Concentrations Assayed)	χ^2^	*p*-Value	Resistance Ratio *
Cry1Ab						
SS-BZ (assayed in 2018)	617	1.41 ± 0.17	0.29 (0.21, 0.42)	132.4	<0.0001	
PPL-Cry1Ab-2018-I	445		>31.6 (33.3%)			>109
PPL-Cry1Ab-2018-II	367	0.75 ± 0.15	3.31(1.99, 6.81)	11.3	0.6620	11
PPL-Cry1Ab-2018-III	402	1.45 ± 0.19	8.6 1(6.34, 12.4)	18.19	0.1099	30
SS-BZ (assayed in 2019)	448	2.10 ± 0.28	0.14 (0.11, 0.19)	22.72	0.0648	
PPL-Cry1Ab-2019	576	1.15 ± 0.16	13.76(9.68, 21.88)	19.13	0.1600	98
Vip3Aa20						
SS-BZ (assayed in 2018)	1190	2.39 ± 0.56	0.36 (0.22, 0.52)	78.4	<0.0001	
PPL-Vip3Aa20-2018-I	375	1.74 ± 0.28	3.96(2.20, 9.51)	6.51	0.0893	11
PPL-Vip3Aa20-2018-II	352	1.60 ± 0.30	2.51(1.54, 4.30)	37.61	0.0006	7
SS-BZ (assayed in 2019)	512	2.64 ± 0.44	0.47 (0.33–0.67)	39.96	0.0007	
PPL-Vip3Aa20-2019	576	1.67 ± 0.16	1.89 (1.46, 2.41)	16.51	0.5567	4

* Resistance ratio was calculated based on the LC_50_ of a strain that was derived from the F_2_ survivors divided by the LC_50_ of the known susceptibility strain, SS-BZ, assayed in the same year.

**Table 3 toxins-14-00270-t003:** Expected genotype frequencies in the F_2_ populations and the expected number of live larvae in the F_2_ screen against Cry1Ab.

Parent Genotype	Expected Genotype Frequency (f) in F_2_ Populations ^a^			Expected Number of Survivors in the F_2_ Screen ^b^
	f_SS_	f_RS_	f_RR_	
SSSS	1	0	0	0
RSSS	0.5625	0.375	0.0625	34.9
RRSS/RSRS	0.25	0.5	0.25	66.5
RRRS	0.0625	0.375	0.5625	94.9
RRRR	0	0	1	120

^a^ f_SS_, f_RS_, and f_RR_ represent the expected SS, RS, and RR genotype frequencies, respectively, in the F2 progeny of an iso-line. ^b^ The expected number of survivors was calculated based on 128 neonates that were screened per iso-line.

**Table 4 toxins-14-00270-t004:** Estimated resistance allele frequencies to Cry1Ab and Vip3Aa20 in *H. zea* populations that were collected from Louisiana and three other southeastern states of the United States.

Bt Toxin	Year	State	Total Iso-Lines Screened	No Resistance Alleles in a Mated Parent Female						Expected Resistance Allele Frequency (95% CI)
				0	1	2	3	4	Total	
				Functional major resistant alleles						
Cry1Ab	2018	Louisiana	32	6	24	2	0	0	28	0.223 (0.156, 0.298)
	2018	Other states	18	2	16	0	0	0	16	0.230 (0.142, 0.331)
	Sub-total 2018		50	8	40	2	0	0	44	0.223 (0.168, 0.283)
	2019	Louisiana	47	2	37	6	2	0	55	0.295 (0.232, 0.361)
	Total	All states	97	10	77	8	2	0	99	0.256 (0.214, 0.301)
Vip3A	2018	Louisiana	33	33	0	0	0	0	0	0 to 0.0218
	2018	Other states	19	19	0	0	0	0	0	0 to 0.0368
	Sub-total 2018		52	52	0	0	0	0	0	0 to 0.0140
	2019	Louisiana	49	49	0	0	0	0	0	0 to 0.0149
	Total	All state	101	101	0	0	0	0	0	0 to 0.0073
Vip3A				Functional minor resistance alleles						
	2018	Louisiana	33	27	6	0	0	0	6	0.054 (0.023, 0.100)
	2018	Other states	19	18	1	0	0	0	1	0.025 (0.003,0.069)
	Sub-total 2018		52	45	7	0	0	0	7	0.039 (0.017, 0.070)
	2019	Louisiana	49	46	3	0	0	0	3	0.020 (0.006, 0.044)
	Total	All states	101	91	10	0	0	0	10	0.028 (0.014, 0.047)

## Data Availability

Not applicable.

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
