# Peer review of "Resistance Allele Frequency to Cry1Ab and Vip3Aa20 in Helicoverpa zea (Boddie) (Lepidoptera: Noctuidae) in Louisiana and Three Other Southeastern U.S. States"

_toxins, 2022, doi:10.3390/toxins14040270_

Round 1
Reviewer 1 Report
Dear Authors,
The research deals with a very interesting, current topic and is treated appropriately. From this derives a clear, fluid, interesting and pleasant to read manuscript. The results are important and will be increasingly important in the near future, indicating that the production strategies of Bt plants will have to undergo a significant variation. The manuscript does not need any revision and as far as my competence is concerned it can be published in this form. line 232 (insert a comma in place of a period). Compliments to the authors.
Author Response
We appreciate the very positive feedback.
Q1. Line 232 (insert a comma in place of a period).
R1. Changed as suggested.
Thanks.
Reviewer 2 Report
This reviewer appreciate the research work did by the authors. Based on the opinion of this reviewer the manuscript could be accepted in the present form. The experimental design and results are well done and support the conclusion indicated in the manuscript
Author Response
We appreciate the very positive feedback.
Reviewer 3 Report
The research works adequately analyze the possible alleles of resistance in H. zea to two Bt toxins, Cry1Ab and Vip3Aa20. The result is very well-founded and structured and presents a cutting-edge topic such as studying the development of resistance in the so-called BT transgenic plants.
- However, it was only missing both in the Introduction and in the Discussion section, commenting more details about the different mechanisms of action of both toxins, explaining why a cross-effect was not observed between their activity at the biological and molecular level and not just restricted to describing the complex data of the resistance alleles.
- Both toxins (Cry1Ab and Vip3Aa20) are expressed at different stages of the life cycle of B. thuringiensis. Was this considered when performing the bioassays?
- Is it known that BT affects neonatal or first instar larvae most effectively. Was the effect of both toxins evaluated in mature larval instars of H. zea?
- Was the joint effect of both toxins evaluated in the same larvae population?
- Were the toxicity levels of both toxins evaluated in each generation obtained?
- It is known that Lepidoptera larvae acquire resistance to Bt toxins, but when the selection pressure is removed, they recover susceptibility to them. Evaluated this beyond an F3, F4, etc??
Author Response
We appreciate the very positive feedback and valuable comments.
Q1. 1. However, it was only missing both in the Introduction and in the Discussion section, commenting more details about the different mechanisms of action of both toxins, explaining why a cross-effect was not observed between their activity at the biological and molecular level and not just restricted to describing the complex data of the resistance alleles.
R1: Bt modes of action are very complicated, and some details are still unknown. However, based on information available, we have known that the Vip3A proteins, compared to the Cry proteins, have different blinding sites in the insect midgut. More detailed information of the Bt blinding molecules has been added in the revised manuscript to address this suggestion (Lines 56-62).
Q2. Both toxins (Cry1Ab and Vip3Aa20) are expressed at different stages of the life cycle of B. thuringiensis. Was this considered when performing the bioassays?
R2. No, the overall bioassay approaches were the same or similar for the two Bt toxins. The bioassays used ‘purified’ Bt proteins produced from Cry1Ab and Vip3Aa20 transgenes, but not from the Bt bacterium itself. Thus, we believe the different production stages of the two proteins by the native Bt bacteria should not be a significant factor to be considered in the bioassay. No further changes for this question were made in the revised manuscript.
Q3. Is it known that BT affects neonatal or first instar larvae most effectively. Was the effect of both toxins evaluated in mature larval instars of H. zea?
R3. Good question! Yes, several studies have envaulted the susceptibilities of different instars to Bt toxins. In most cases, the first instars are usually more susceptible than late instars. In the crop fields planted with transgenic Bt corn, Bt proteins are usually expressed throughout the entire plant stages and thus in most cases, neonates will first encounter the Bt toxins expressed by the plants and the exposures to the Bt toxin will continue if the neonates (larvae) are not killed. Varied Bt susceptibilities in different larval stages plus other factors such as larval movement and seed blend planting might cause a ‘high dose’ to become a ‘non-high dose’ for resistance management. However, this issue is beyond the scope of the current study. No changes for this question were made in the revised manuscript.
Q4. Was the joint effect of both toxins evaluated in the same larvae population?
R4. No, not yet. The objective of the current study was to determine the resistance allele frequencies to individual Bt toxins. Synergism or antagonism of two Bt proteins could play a significant role for survival of the insects on Bt plants containing two or more Bt genes. However, this issue is also somewhat beyond the scope of the current student. We modified little bit of the text in the section of discussion and added a related reference to address this question (Lines 351, and 702-704).
Q5. Were the toxicity levels of both toxins evaluated in each generation obtained?
R5. As described in the materials and methods. The diet overlay bioassays were performed in two generations for SS-BZ and for each of the two toxins, while only one generation for other insect strains. The goal of the bioassays was to confirm if the PPL strains were resistant to the Bt toxins. No additional changes for this question were made in the revised manuscript.
Q6. It is known that Lepidoptera larvae acquire resistance to Bt toxins, but when the selection pressure is removed, they recover susceptibility to them. Evaluated this beyond an F3, F4, etc??
R6. This is an issue related to fitness costs of resistance. There is still a lot of work (such as more selection to generate homozygous resistant strains, backcrosses to ensure similar genetic background among insect strains to be tested, crosses to generate F1 hybrids, ….) that must be done before these PPL strains should be used to evaluate fitness costs of the resistance. As discussed above, the diet overlay bioassays of the current study were only for resistance confirmation of these PPL strains. Thus, no additional changes for this question were made in the revised manuscript.
Thanks.
Reviewer 4 Report
The authors presented a very well written manuscript. The resistance monitoring is very important to avoid future problems with the use of Bt technology.
The introduction, material and methods and results parts are well written. The discussion section is well structured and debate the results sufficiently.
I have few suggestions to give:
Line 36: Please, provide the scientific names of the plants. Maize: Zea mays L. and cotton: Gossypium hirsutum L.
Line 39: Which countries? Please, could you cite them?
Line 47: U.S.: United States
Table 1: The first and second columns of the table is confusing. Maybe adding a line between the "Sub-total (2018)" and "2019 Louisiana" would help turn the table more easy to understand.
Table 3: Please, describe the meaning of fSS, fRS, fRR.
Table 4: U.S.: United States

Author Response
Response: We appreciate the very positive feedback.
Q1. Line 36: Please, provide the scientific names of the plants. Maize: Zea mays L. and cotton: Gossypium hirsutum L.
R1. The scientific names have been added as suggested.
Q2. Line 39: Which countries? Please, could you cite them?
R2. Names of the sic countries have been added as suggested.
Q3. Line 47: U.S.: United States
R3. Changed as suggested.
Q4. Table 1: The first and second columns of the table is confusing. Maybe adding a line between the "Sub-total (2018)" and "2019 Louisiana" would help turn the table more easy to understand.
R4. Modified as suggested
Q5. Table 3: Please, describe the meaning of fSS, fRS, fRR
R5. Information has been added as suggested
Q6. Table 4: U.S.: United States.
R6. Changed as suggested.
Thanks.